

**Seasonal Air Concentration Variability, Gas/Particle Partitioning,**
**Precipitation Scavenging, and Air-Water Equilibrium of**
**Organophosphate Esters in Southern Canada**
Yuening Li,[1] Faqiang Zhan,[1] Chubashini Shunthirasingham,[2] Ying Duan Lei,[1] Jenny Oh,[1,3]
Amina Ben Chaaben,[4] Zhe Lu,[4] Kelsey Lee,[5] Frank A. P. C. Gobas,[5] Hayley Hung,[2] Frank
Wania[1,3*]
*[1] Department of Physical and Environmental Sciences, University of Toronto Scarborough,*
*1265 Military Trail, Toronto, Ontario, Canada M1C 1A4*
*[2] Environment and Climate Change Canada, Downsview, 4905 Dufferin St, North York,*
*Ontario, Canada M3H 5T4*
*[3] Department of Chemistry, University of Toronto Scarborough, 1265 Military Trail, Toronto,*
*Ontario, Canada M1C 1A4*
*[4] Institut des Sciences de la Mer de Rimouski, Université du Quebec à Rimouski, 300 allée*
*des Ursulines, Rimouski, Québec, Canada G5L 3A1*
*[5] School of Resource and Environmental Management, Simon Fraser University, 8888*
*University Dr, Burnaby, British Columbia, Canada V5A 1S6*
*Corresponding author: frank.wania@utoronto.ca





**Abstract**
In response to increasing production and application volumes, organophosphate esters
(OPEs) have emerged as pervasively detected contaminants in various environmental media,
with concentrations often exceeding those of traditional organic contaminants. Despite the
recognition of the atmosphere's important role in dispersing OPEs and a substantial number
of studies quantifying OPEs in air, investigations into atmospheric phase distribution
processes are rare. Using measurements of OPEs in the atmospheric gas and particle phase,
in precipitation and in surface water collected in Southern Canada, we explored the seasonal
concentration variability, gas/particle partitioning behaviour, precipitation scavenging, and
air-water equilibrium status of OPEs. Whereas consistent seasonal trends were not observed
for OPEs concentrations in precipitation or atmospheric particles, gas phase concentrations
of several OPEs were elevated during the summer in suburban Toronto and at two remote
sites on Canada's east and west coast. Apparent enthalpies of air-surface exchange fell
mainly within or slightly above the range of air/water and air/octanol enthalpies of exchange,
indicating the influence of local air-surface exchange processes and/or seasonally variable
source strength. While many OPEs were present in notable fraction in both gas and particle
phase, no clear relationship with compound volatility was apparent, although there was a
tendency for higher particle-bound fractions at lower temperature. High precipitation
scavenging ratios for OPEs measured at the two coastal sites are consistent with low air-
water partitioning ratios and the association with particles. Although beset by large
uncertainties, air-water equilibrium calculations suggest net deposition of gaseous OPEs
from the atmosphere to the Salish Sea and the St. Lawrence River and Estuary. The
measured seasonal concentration variability is likely less a reflection of temperature driven
air-surface exchange and instead indicates that more OPE enter, or are formed in, the
atmosphere in summer. More research is needed to better understand the atmospheric gas-
particle partitioning behaviour of the OPEs and how it may be influenced by transformation
reactions.
**Key words:**
OPEs, air, precipitation, water, partitioning, air-water exchange, relative abundance



## 1. INTRODUCTION

Organophosphate esters (OPEs) are synthetic organophosphorus compounds consisting of a central phosphate molecule substituted with non-halogenated, halogenated alkyl, or aryl groups. Widely used as flame retardants, plasticizers, stabilizers, and defoaming agents in various industries and consumer products (Environment and Climate Change Canada, 2023abcd; Salamova et al., 2016; van der Veen and de Boer, 2012), OPEs are typically physically incorporated into materials rather than chemically bonded (Wang et al., 2020b; Wong et al., 2018), facilitating their release into the environment. Following restrictions on many brominated flame retardants, e.g. through listing in the Stockholm Convention, OPEs use has increased, reaching 620 kilotons globally in 2013, accounting for 30% of total flame retardant usage (Sühring et al., 2016; Xie et al., 2022). The extensive application of OPEs, coupled with their potential for long-range atmospheric transport (Na et al., 2020; Sühring et al., 2016) and persistence (Möller et al., 2012; Salamova et al., 2014), has resulted in their ubiquitous presence in the environment (Han et al., 2020; Li et al., 2019a, b; Lu et al., 2017; Mi et al., 2023; Regnery and Püttmann, 2009; Stackelberg et al., 2007), often at concentrations exceeding those of traditional flame retardants and plasticizers (Salamova et al., 2014; Shoeib et al., 2014; Zhao et al., 2021b). Given their potential toxicity (Gu et al., 2019; Li et al., 2020; Rosenmai et al., 2021; Wang et al., 2022; Yan and Hales, 2019, 2020), understanding the fate, occurrence, and distribution of OPEs in the environment is critical for assessing their ecological and human health impacts.

The atmosphere plays a key role in the dispersion and transport of OPEs, with concentrations and spatial and temporal variability in air being influenced by emission sources, atmospheric transport, chemical transformation (Liu et al., 2023; Liu and Mabury, 2019) and deposition processes. The distribution of OPEs between different atmospheric phases (gas phase, particles, precipitation) affects these processes and is influenced by their partition properties. Most studies on OPEs in the atmosphere report concentrations in the particle phase, whereas studies on the presence in the gas phase are far more limited, which may be related to the relatively short half-lives of gas phase OPEs (Shi et al., 2024; Zhang et al., 2016). However, gaseous OPEs can constitute 15% to 65% of atmospheric OPEs (Möller et al., 2011), and diffusive air-water gas exchange of OPEs can be 2-3 orders of magnitude higher than dry particle deposition (Castro-Jiménez et al., 2016; Ma et al., 2021), highlighting the need for more research on OPE vapours.

Precipitation acts as a major pathway for the removal and redistribution of OPEs from the



atmosphere to aquatic and terrestrial environments (Shi et al., 2024). It can scavenge and
deposit both gas-phase and particle-bound OPEs. Depending on regional emissions,
temperature, precipitation type, and the physicochemical properties of the OPEs (Lei and
Wania, 2004), the wet deposition flux of OPEs can be significantly larger than the dry
deposition flux (Kim and Kannan, 2018). Despite its importance, fewer than ten studies have
reported OPE concentrations in precipitation (Bacaloni et al., 2008; Casas et al., 2021; Fries
and Püttmann, 2003; Kim and Kannan, 2018; Marklund et al., 2005b; Mihajlović and Fries,
2012; Regnery and Püttmann, 2009; Zhang et al., 2020), and only one study has reported
precipitation scavenging ratios for atmospheric OPEs (Casas et al., 2021).
OPEs can enter water bodies through air-water gas exchange (Castro-Jiménez et al., 2016;
Ma et al., 2021), wet and dry deposition (Castro-Jiménez et al., 2016; Kim and Kannan,
2018; Ma et al., 2021), wastewater effluent (Marklund et al., 2005a), industrial and
municipal discharges (Bacaloni et al., 2008; Fries and Püttmann, 2003), and surface runoff
(Awonaike et al., 2021; Regnery and Püttmann, 2010). Some OPEs, including tris(1-chloro-
2-propyl) phosphate (TCPP) and tris (phenyl) phosphate (TPhP), have been detected in fish
(Ma et al., 2013; Sundkvist et al., 2010). A comprehensive understanding of the
environmental fate and occurrence of OPEs, and in particular a better understanding of the
contribution that the atmosphere makes for the delivery of OPEs to aquatic ecosystems,
would benefit from investigations that quantify OPE concentrations in multiple
environmental media sampled in the same area and at the same time. Despite the substantial
number of studies on OPEs in the environment, those examining OPEs across three or more
phases are very rare (He et al., 2019; Li et al., 2019b; Mi et al., 2023) and most studies focus
on just one or two media, usually gas and/or particle phases (Li et al., 2018; Ma et al., 2022;
Sühring et al., 2016; Zhao et al., 2021b) or the water phase (Choo and Oh, 2020; Ding et al.,
2015; McDonough et al., 2018; Shi et al., 2020). No previous study has investigated OPEs
in atmospheric gas and particle phases, precipitation, and surface water simultaneously.
To address this research gap and gather information on the contribution that the atmosphere
makes to OPEs in coastal waters of Southern Canada, we aimed to characterize the
occurrence, behavior, and fate of OPEs in different atmospheric phases. We measured OPEs
in precipitation and atmospheric gas and particle phase for one year at two remote sites on
Canada's East and West coast, respectively, and complemented this dataset with the results
of a year-long measurement campaign of OPEs in the gas and particle phase in Toronto. We
further used passive samplers to gather data on the spatial variability of OPE concentrations



in the atmospheric gas phase and in water in the two coastal regions. The passive air
sampling data have been presented previously (Li et al., submitted). This unique data set
allowed us to estimate the gas-particle distribution in the atmosphere, precipitation
scavenging ratios, and the state of air-surface water equilibrium, often in their seasonal
dependence or their variability between urban, rural and remote locales. Finally, we used
this dataset to explore the relative abundance of OPEs in the different types of samples.
**2. MATERIALS AND METHODS**
**2.1 Active Air Sampling and Precipitation Collection.** 24-hour air samples were collected
monthly for one year using a high-volume active air sampler (AAS); twelve at a location on
Saturna Island, British Columbia (BC) (48.7753N, -123.1283W), and twelve in the vicinity
of Tadoussac, Quebec (QC) (48.1415N, -69.6991W). Forty-eight consecutive week-long
AASs were taken with a mid-volume pump in the Eastern suburbs of Toronto (43.78371 N,
−79.19027 W) (Li et al., 2023a, b, 2024). At all three sites, polyurethane foam
(PUF)/XAD/PUF sandwiches and glass-fiber filters (GFFs) were used to collect OPEs in
the gas and particle phase, respectively. Precipitation samples (PCPNs) were collected at the
AAS sampling locations in BC and QC during the same months as the air samples and the
sampling length was ~ 30 days (Oh et al., 2023; Zhan et al., 2023).
**2.2 Passive Air and Water Sampling.** In QC, 86 passive air samplers (PASs) were
deployed at 71 unique sampling sites on either shore of the St. Lawrence River and Estuary,
including in Montreal and Quebec City between 2019 and 2022. In BC, 83 PASs were
deployed at 47 sites in the lower mainland around Vancouver and on the Canadian shore of
the Salish Sea during different time periods between 2020 and 2022. More details are given
in Table S5 in the Supporting Information (SI) of Li et al. (submitted) and in previous
publications (Oh et al., 2023; Zhan et al., 2023).
Forty-eight low-density polyethylene (LDPE) based passive water samplers (PWSs) were
spiked with performance reference compounds (PRCs), deployed at 10 sites in BC and 10
sites in QC, and collected after deployment lasting 20-35 days in BC and 27-70 days in QC.
Detailed information on the PWS sampling is provided in the SI (Table S13) and previous
publications (Oh et al., 2023; Zhan et al., 2023).
**2.3 Sample Analysis.** Prior to extraction all samples were spiked with seven isotopically
labeled OPEs (Table S1) as surrogates. XAD from the PASs, the PUF/XAD sandwiches and
GFFs from the AASs were extracted using a Dionex Accelerated Solvent Extractor 350. The





PCPN and PWS samples were extracted using liquid-liquid extraction with dichloromethane
and soaking in hexane, respectively. Extracts were concentrated to 0.5 mL using a rotary
evaporator and nitrogen blow-down. Triamyl phosphate was added into the concentrated
extracts as an injection standard. Gas chromatography-tandem mass spectrometry (GC-
MS/MS) was used to detect and quantify 16 OPEs (Tables S1 and S2).
**2.4 Quality Assurance and Quality Control.** All extraction and concentration procedures
were carried out in a trace analytical laboratory. The glassware was cleaned using a machine
with detergents, then rinsed with deionized water, and finally baked with GFFs at 450 °C in
a muffle furnace for 24 hours. Experimental materials that came into contact with samples
or extracts were thoroughly cleaned and rinsed three times with solvents (acetone and
hexane, or dichloromethane) before use. Field blanks, procedure blanks, and solvent blanks
were prepared with each batch of extractions and analyses (Oh et al., 2023; Zhan et al., 2023).
OPEs were not found in procedure or solvent blanks. Only a few analytes were present in
the field blanks, and for these, the average detected amount was subtracted from the amounts
of target chemicals in the field samples. Method detection limits (MDLs) were calculated as
three times the standard deviations of levels in field blanks when analytes were detected
(signal-to-noise ratio (S/N) > 3); otherwise, MDLs were based on concentrations at which
S/N is 3 (Desimoni and Brunetti, 2015). MDLs are provided in the Supplementary
Information (Tables S5, S8, S10, S11, and S13). The average recoveries of five surrogates
in AASs, PCPNs, and PWSs ranged from 78% to 232% (Table S3). The concentrations
reported have been corrected for recovery.
**2.5 Data Analysis.** Water concentration of OPEs were calculated from the amounts
quantified in PWS extracts following the method by Booij and Smedes (Booij et al., 2003;
Booij and Smedes, 2010), with details provided by Oh et al. (2023).
The fraction of an OPE in the particle phase ($\Phi$, %) was obtained by dividing the particle-
phase concentration by the sum of concentrations in the gas and particle phase. Gas-particle
partition ratios $K_{PA}$ (m$^3$ air g$^{-1}$ aerosol) were derived by dividing the measured concentrations
of an OPE in the particle phase (pg m$^{-3}$) by the product of the concentrations of particles less
than 2.5 µm in diameter (PM$_{2.5}$, g m$^{-3}$) obtained from nearby national air pollution
surveillance program (NAPS) stations (Table S8) and the measured concentrations of this
OPE in the gas phase (pg m$^{-3}$). More detail is given in previous publications (Li et al., 2023a;
Oh et al., 2023; Zhan et al., 2023).



Measured scavenging ratios (SRs) were calculated as the ratios between the concentrations
of an OPE in precipitation and air (sum of gas and particle phase). We also estimated SRs
by assuming equilibrium of OPE between the atmospheric gas phase and water droplets (Oh
et al., 2023), and that all OPEs are sorbed to the same particles, which are scavenged with a
scavenging ratio $W_P$ of 200,000 (Kim et al., 2006). An estimated SR thus is $(1-\Phi)K_{WA}$ +
$\Phi W_P$, where $K_{WA}$ is the temperature-adjusted partition ratio between water and air ($K_{WA} = $
$K_{AW}^{-1}$, Table S4).
The fugacities of OPEs in water $f_W$, at average sea surface temperature $T_W$ in K, were
calculated using $C_W \cdot K_{AW}(T_W) \cdot R \cdot T_W$, and those in air ($f_A$), at average air temperature $T_A$ in
K, were derived with $C_A \cdot R \cdot T_A$, where $C_W$ and $C_A$ are the OPE concentrations (mol m$^{-3}$) in
water and air, respectively, and $R$ is the gas constant.
**3. RESULTS**
**3.1 OPEs in the Atmospheric Gas Phase.** The gas phase concentrations obtained during
the three one-year AAS campaigns in Tadoussac, on Saturna Island, and in Toronto are given
in Table S5. The gas phase concentrations obtained by passive air sampling in QC and BC
have been previously reported (Li et al., submitted) with tri-n-butyl phosphate (TBP),
tris(2-chloroethyl) phosphate (TCEP), tris(1-chloro-2-propyl) phosphate (TCPP), and tris
(phenyl) phosphate (TPhP) being reliably and ubiquitously detected. Due to the higher
sampling volumes of the AAS (~520 m$^3$) compared to the PAS (less than 200 m$^3$), more
OPEs could be detected above the MDL in the AAS. At all three locations, TBP, TCEP,
TCPP, TPhP, and 2-ethylhexyl-diphenyl phosphate (EHDPP) were present above the MDL.
Additionally, triethyl phosphate (TEP) was detected on Saturna Island, TEP,
tris(1,3-dichloro-2-propyl) phosphate (TDCPP), and tris (2-butoxyethyl) phosphate (TBEP)
were detected in Tadoussac, and tri-propyl phosphate (TPrP) and TDCPP were detected in
Toronto. We are no comparing here the gas phase concentrations recorded in our study with
those reported previously, because that had already been done in Li et al. (submitted)
For the four most frequently detected OPEs, it is possible to compare the levels obtained
with the AASs on Saturna Island and in Tadoussac and by PASs at the nearby sites L43 and
S57. On Saturna Island, the PAS deployment at site L43 overlapped with the timeframe of
the AASs (between May and October 2020) (Table S8). In Tadoussac, the deployment
period of the PAS at S57 (November 2019 – August 2020) preceded the AASs sampling by
about one year (December 2020 – September 2021). Except for TBP and TPhP on Saturna



Island, PAS levels generally trended lower than AAS levels at both locations, albeit within
a factor of 5. One contributing factor to this difference could be the episodic 24-hour active
air sampling's inability to represent long-term concentration levels compared to PAS. For
instance, AAS-measured concentrations of TBP in Tadoussac ranged from below detection
to approximately 200 pg m$^{-3}$. Another factor could be the spatial distribution variability of
atmospheric OPEs. Despite our efforts to use PAS data from sites closest to AAS locations
for comparison, the PAS and AAS sampling sites were not identical. To support the
hypothesis that spatial and temporal variability in OPE concentrations contributes to the
discrepancy, we also compared AAS and PAS results for hexachlorobutadiene (HCBD) and
hexachlorobenzene (HCB), which exhibit uniform spatial distribution and consistent
concentrations over time, using the same samples as for the OPEs. PAS levels for these two
compounds closely aligned with AAS levels within a factor of 1.5. Similarly,
halomethoxybenzene levels from PASs and AASs were within a factor of 3 (Zhan et al.,
222    2023).

**3.2 OPEs in Atmospheric Particle Phase.** The concentrations of five OPEs (TBP, TCEP,
TCPP, TPhP, EHDPP) in the atmospheric particles from the three AAS sampling locations
are compiled in Tables S9. Except for TPhP and EHDPP which were not detected in particles
from Saturna Island, all five OPEs most frequently detected in the gas phase could also be
quantified in particle samples. Again, TCPP is the most abundant OPE at all three sites.
Concentration levels on Saturna Island and in suburban Toronto are similar and almost one
order of magnitude higher than those in Tadoussac. The averaged TBP levels of 7 pg m$^{-3}$, 3
pg m$^{-3}$, and 9 pg m$^{-3}$ on Saturna Island, in Tadoussac, and in Toronto are lower than those
in Antarctica (23 pg m$^{-3}$) (Wang et al., 2020a) and two order of magnitude lower than those
detected in cities in the Great Lakes area (130 pg m$^{-3}$) in 2012 (Salamova et al., 2013).
Except in Tadoussac (1 pg m$^{-3}$), the TCEP levels of 50 pg m$^{-3}$ and 17 pg m$^{-3}$ on Saturna
Island and in Toronto are higher than those in Antarctica (5 pg m$^{-3}$) (Wang et al., 2020a) and
ca. 2~4 times lower than those detected in cities in the Great Lakes region (89 pg m$^{-3}$)
(Salamova et al., 2013), and two orders of magnitude lower than the reported median
concentration in Quebec City and near the St Lawrence River (1903 pg m$^{-3}$) (Sühring et al.,
2016). TCPP in Tadoussac, 3 pg m$^{-3}$, is comparable to its level in Antarctica (6 pg m$^{-3}$)
(Wang et al., 2020a), and TCPP on Saturna Island (122 pg m$^{-3}$) and in Toronto (90 pg m$^{-3}$)
are three and four times lower than those detected in cities in Great Lakes area (321 pg m$^{-3}$)
(Salamova et al., 2013), and one order of magnitude lower than the detected level in Quebec



City and near the St Lawrence River (1557 pg m$^{-3}$) (Sühring et al., 2016). TPhP and EHDPP
levels in Antarctica (1 pg m$^{-3}$) (Wang et al., 2020a) are close to those in Tadoussac (2 pg m$^{-3}$),
and one order of magnitude lower than levels in Toronto (12 pg m$^{-3}$) and those in Quebec
City and near the St Lawrence River (51 pg m$^{-3}$) (Sühring et al., 2016). The relatively higher
concentration levels of certain OPEs in Antarctica, such as TBP, may be due to preferential
partitioning of TBP to particles at low temperatures. Compared to the sites in the Great Lakes
region (Salamova et al., 2013) as well as Quebec City and near the St Lawrence River region
(Sühring et al., 2016), our sampling sites were more rural, which could explain lower OPE
concentrations.
**3.3 OPEs in Precipitation.** Eight OPEs, i.e., TEP, TBP, TCEP, TCPP, TDCPP, TPhP,
TBEP, and EHDPP, were reliably detected in the precipitation samples from Saturna Island
and Tadoussac (Table 1 & Table S11). Concentrations are generally higher on Saturna Island
than in Tadoussac. The OPE levels in Tadoussac were comparable to those in Antarctica
(Casas et al., 2021). TDCPP detected on Saturna Island and in Tadoussac are two times to
one order of magnitude higher than the levels in Antarctica (Casas et al., 2021), Nanjing
(Zhang et al., 2020), and New York (Kim and Kannan, 2018), and our measured EHDPP
levels were higher than those detected in Antarctica. Overall, except for TDCPP and EHDPP,
the average OPE concentrations detected in our study were comparable or one order of
magnitude lower than literature data (Table 1). Except for TBP in Tadoussac, OPE
concentrations varied greatly between months, whereby no distinct and consistent seasonal
trends were discernible (Table S11), which is consistent with previous observations
(Regnery and Püttmann, 2009).
**3.4 OPEs in Water.** The OPEs concentrations in water, obtained with PWSs deployed in
the summer 2021, are reported in Table S13. Their spatial patterns are displayed in Figures
1, S1, and S2. In BC, OPEs had elevated levels in the interior of Burrard Inlet close to Port
Moody (V1 and V2), at the southern mouth of the Fraser River (V5), and at some sites
around populated areas in Victoria, BC (V6-V8). TBP, TCPP, and TDCPP had higher
concentrations close to an industrial area near Esquimalt (V10). In QC, highest OPE water
concentrations were usually detected at site W5, in the Saint Lawrence River close to an
industrial area in Sorel-Tracy, rather than at sites in Montreal (W1 and W2) or Québec City
(W8 and W9). W4 also had elevated concentrations for some OPEs such as TBP. Water
concentrations at the one sampling site in the Saint Lawrence Estuary were much lower than
in the river. Overall the spatial patterns suggest that the water concentrations of OPEs were





275 related to both industrial activities and human populations in BC, whereas industrial
276 activities might have relatively higher impact on water concentrations of OPEs in QC. It
277 should be noted that the dispersion plume of the Montreal waste water treatment plant enters
278 the river at 45 40' N, 73 28' W and stays on the north side of the river (Marcogliese et al.,
279 2015), therefore, the OPEs in the dispersion plume might not be sampled at W3 and W4.

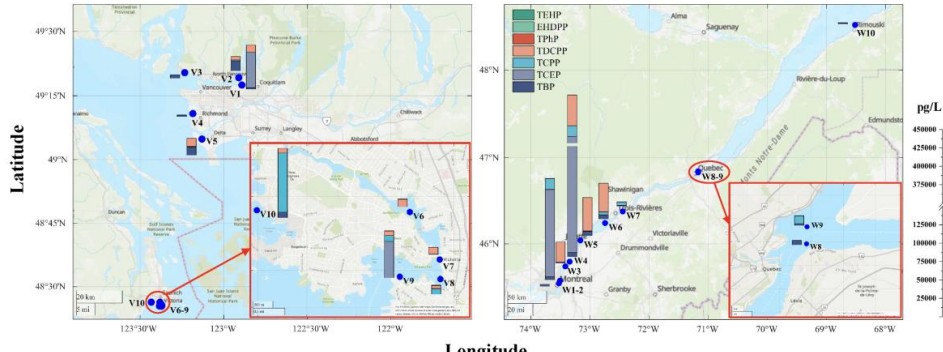

280

281 **Figure 1** Spatial patterns of OPEs in the water in British Columbia (left panel) and Quebec (right
282  panel). The inserted maps at the bottom right of each panel show the sampling sites
283  located within Victoria (left panel) and Quebec City (right panel). Concentrations in
284  duplicate samples were averaged. The stacked bars indicate the total concentrations
285  levels of all detected OPEs and individual OPE. Various colors are used for different
286  OPEs. The concentration scale is shown to the right of the maps, which were created
287  using the basemap of MATLAB, copyrighted to Esri, TomTom, Garmin, SafeGraph,
288  GeoTechnologies, Inc, METI/NASA, USGS, Bureau of Land Management, EPA, NPS,
289  US Census Bureau, USDA, USFWS, NRCan, and Parks Canada.

290 **4. DISCUSSION**

291 **4.1 Seasonality and Temperature Dependence.** Clear and consistent seasonal trends were
292 not observed for particle-bound OPEs at any location during the one year of sampling
293 (Tables S9). However, OPE gas phase concentrations at all three AAS sampling sites clearly
294 varied seasonally, allowing us to investigate the influence of temperature on those
295 concentrations (Figure 2). Except for EHDPP and TDCPP, concentrations of frequently
296 detected OPEs increased with increasing ambient temperature. The logarithm of the partial
297 pressures of OPEs (ln $p$) were linearly regressed against the reciprocal of absolute
298 temperatures ($1/T$) (Clausius-Clapeyron (CC) relationship), with the slopes, $R^2$ values, and
299 $p$ values summarized in Table S7. Regressions for TBP were significant at the three AAS
300 sites ($p < 0.05$), whereas EHDPP did not show significant relationships at any site. The CC



relationships for other OPEs were only significant ($p$ <0.05) at some locations, i.e., TCEP
and TCPP on Saturna Island, TEP and TPhP in Tadoussac, and TPrP, TCPP, TCEP, and
TDCPP in Toronto. In cases with $R^2$ >0.10, the trends indicate higher partial pressures at
higher temperatures.
Apparent enthalpies of air-surface exchange ($\Delta H_{\text{AS-app}}$) were obtained from the slopes of the
CC relationships with $R^2$ >0.30 and $p$ <0.05, and compared with enthalpies of exchange
between air and water ($\Delta H_{\text{AW}}$) and between air and octanol ($\Delta H_{\text{AO}}$), estimated using poly-
parameter linear free energy relationships (UFZ-LSER database v 3.2.1 [Internet], 2024)
(Table S7). Values of $\Delta H_{\text{AS-app}}$ that are similar to $\Delta H_{\text{AW}}$ and $\Delta H_{\text{AO}}$ have been interpreted as
being indicative of a dominant contribution of temperature-driven local air-surface
exchanges on the air concentration at a site (Bidleman et al., 2023; Wania et al., 1998; Zhan
et al., 2023). If $\Delta H_{\text{AS-app}}$ is much lower than $\Delta H_{\text{AW}}$ and $\Delta H_{\text{AO}}$, advection from elsewhere is
presumed to play a more important role. $\Delta H_{\text{AS-app}}$ values of OPEs at the three sampling sites
were mostly within the uncertainty range of $\Delta H_{\text{AW}}$ and $\Delta H_{\text{AO}}$. In several instances the
temperature dependence of air concentrations was even larger than might be expected from
air-surface equilibrium, i.e. $\Delta H_{\text{AS-app}}$ was larger than $\Delta H_{\text{AW}}$ and $\Delta H_{\text{AO}}$. Examples are the
$\Delta H_{\text{AS-app}}$ values of TCPP on Saturna Island and in Toronto, as well as those of TBP and
TPhP in Tadoussac, and, to a smaller extent also TCEP on Saturna Island and in Toronto,
and TEP in Tadoussac.
This may simply be a result of high uncertainty, considering the relatively small number of
samples available for deriving the CC relationships for Saturna Island and Tadoussac. It
could also suggest that temperature influences not only the exchange between air and surface
but also the OPE source strength to the atmosphere. This source strength could be correlated
with temperature, e.g., because of enhanced release of OPEs from materials at higher
temperatures or higher indoor-outdoor exchange rates in summer. Furthermore, the
formation of TCPP, TCEP, and TPhP from precursor compounds (i.e., tris(2-
chloroisopropyl) phosphite (TCPPi), tris(2-chloroethyl) phosphite (TCEPi), and triphenyl
phosphite (TPhPi) by reaction with ozone  could be higher in summer (Liu et al., 2023; Liu
and Mabury, 2019; Turygin et al., 2018; Zhang et al., 2021), when photooxidant
concentrations tend to be higher. Even though TCPP is widely used in large quantities, the
spatial distribution and usage of its precursor TCPPi has not been reported. The high $\Delta H_{\text{AS-}}$
$_{\text{app}}$ of TEP and TBP in Tadoussac may also be related to the conversion of their
corresponding phosphite esters.



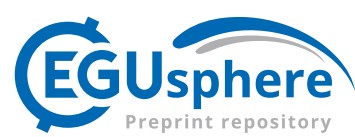

**Table 1** Summary of levels OPEs in precipitation reported in the literature and our study.

| Region & locations | Years | Concentration, ng/L | | | | | | | | References | Note |
|---|---|---|---|---|---|---|---|---|---|---|---|
| | | TEP | TBP | TCEP | TCPP | TDCPP | TPhP | TBEP | EHDPP | | |
| **Literature data** | | | | | | | | | | | |
| Livingston Island, Antarctic | 2018 | 2.1 | 1.0 | 3.1 | 26.0 | 1.9 | | | 0.11 | (Casas et al., 2021) | |
| Nanning, China | N/A | | 4.0 | 15 | 38 | 2.1 | 1.0 | | | (Zhang et al., 2020) | Mean |
| Osnabrueck, Germany | 2011 | | | 187 | 372 | 46 | | | | (Mihajlović and Fries, 2012) | Median |
| Bahnbrücke, Germany | 2001 | | 911 | 121 | | | | 394 | | (Fries and Püttmann, 2003) | |
| Rome, Italy | 2007 | 46 | 46 | 155 | 686 | 404 | | 112 | | (Bacaloni et al., 2008) | Mean |
| Martignano, Italy | 2007 | 12 | 11 | 19 | 28 | 108 | | 38 | | (Bacaloni et al., 2008) | |
| New York, USA | 2017 | 17.7 | 3.9 | 5.7 | 61.8 | 11.7 | 11.0 | | | (Kim and Kannan, 2018) | Mean |
| **Our study** | | | | | | | | | | | |
| Saturna Island | 2020 | 3.0 | 4.0 | 15.6 | 25.7 | 20.9 | 1.0 | 13.7 | 1.2 | Our study | Mean |
| Tadoussac | 2021 | 1.2 | 0.6 | 2.6 | 5.1 | 37.0 | 0.6 | 2.8 | 0.5 | Our study | Mean |

The concentrations of OPEs in snow and rain water samples from five locations in Germany during 2007-2008 were reported (Regnery and Püttmann, 2009). However, as we could not calculate the average OPE concentrations in precipitation, we did not include these data in this table.
Marklund et al.(2005b) reported the concentrations of OPEs in combined dry and wet deposition samples, as there are no data for precipitation samples, therefore, these data were not included in this table as well.





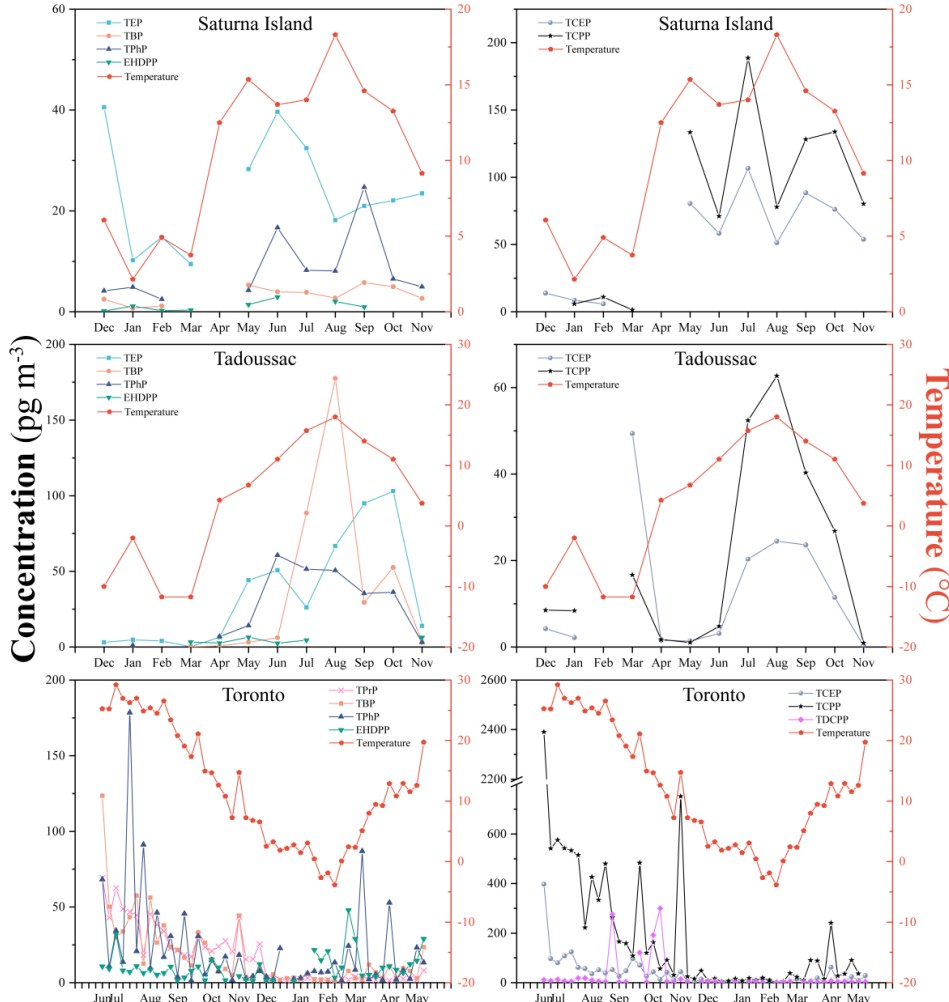

**Figure 2**     Seasonal variability in ambient temperature (right y axis) and gas phase concentrations of OPEs (left y axis) in the atmosphere of Saturna Island, BC (2019-2020, top), Tadoussac, QC (2020-2021, middle), and Toronto, Ontario (2020-2021, bottom). Only data for OPEs with detection frequency higher than 50% are shown.

Incidentally, at 11 sites in BC, where PASs were deployed at least three times during different seasons with different average temperatures, higher OPEs concentrations were generally also observed during warmer deployments (Tables S6 and Table S5 in the SI of Li et al. submitted). 33 out of 55 CC relationships using these PAS data had $R^2$ >0.5, and 27 of these 33 were negative (Table S6). Considering the limited number of data points (3~4) for PAS sites with multiple deployments in different seasons, $\Delta H_{\text{AS-app}}$ values may have high





uncertainties (Table S6) and were therefore not compared with theoretical values.
**4.2 Gas-Particle Partitioning.** The fraction of the OPEs in the particle phase ($\Phi$, %) are
given in Table S9. As more than 50% of $\Phi$ values for TPhP and EHDPP in Tadoussac were
calculated using values < MDL, these data are not discussed further. Overall, $\Phi$ ranged
between 32% and 68 % and varied between OPEs and location. Among the five OPEs, the
$\Phi$ of TCEP is the smallest at almost all three AAS sites. The $\Phi$ values for TBP and TCPP
on Saturna Island (both ca. 66%) are 12 % higher than that for TCEP (54%). In Tadoussac,
the $\Phi$ for TCPP (38%) is comparable to that for TCEP (34%), whereas $\Phi$ for TBP (52%) is
the highest among three OPEs. In Toronto, the $\Phi$ values of the five major OPEs were in the
sequence of TCEP (50%) = TBP (50%) < TPhP (54%) < TCPP (56%) < EHDPP (68%).
This sequence is opposite to that found above the North Atlantic Ocean and in the Arctic
(Wu et al., 2020).
Theoretically, TCEP, TCPP, and TBP have very similar volatility with logarithmic
equilibrium partition ratios between octanol and air (log $K_{OA}$) around 9 and log ($K_{PA}$ / m$^3$ g$^{-1}$
$^1$) of ~1 at 15 °C estimated using the UFZ-LSER website (UFZ-LSER database v 3.2.1, 2024)
(Table S9). These three chemicals are expected to be largely in the gas phase at ambient
temperatures. TPhP and EHDPP have estimated log $K_{OA}$ values > 12 and log ($K_{PA}$ / m$^3$ g$^{-1}$)
of ~5 at 15 °C which would indicate strong particle sorption in the atmosphere. However,
the unexpectedly low fraction observed in the particle phase may suggest that TPhP and
EHDPP are emitted at higher temperatures and are not in a state of equilibrium between gas
and particle phase (Zhao et al., 2021a). The composition of the particles, relative humidity
(Li et al., 2017; Wu et al., 2020), and degradation of OPEs in gas and particle phases may
also influence the gas-particle partitioning of OPEs.
The calculated $\Phi$ at the three AAS sites increases with decreasing ambient temperatures.
This is consistent with lower temperatures favoring partitioning to particles (Table S9). This
is also reflected in the positive linear relationships between the ln $K_{PA}$ and reciprocal
temperature (in K) in Tadoussac, Saturna Island, and Toronto (Table S10).
**4.3 Scavenging Ratios.** Measured SRs could be calculated for eight OPEs and ranged
mainly from $10^4$ to $10^7$ (Table S12). These SRs are highly uncertain because of the
uncertainty in the measured concentrations and because we combine a monthly precipitation
sample with a 24-hour air sample taken during the same month. The estimated SRs are also
uncertain due to the possibly high uncertainty in the estimated $K_{WA}$ and the assumptions





regarding equilibrium partitioning of OPE vapors between air and water droplets and the
value and constancy of $W_P$. Despite these uncertainties, estimated SRs for TBP and EHDPP
are generally around $10^5$ and therefore comparable to the measured ones, which indicates
that equilibrium between precipitation and these chemicals in the atmosphere was achieved.
The estimated SRs for other OPEs are mostly within the range of $2\times10^6$ to $10^9$ and therefore
orders of magnitude higher than the measured SRs. At very high values, exceeding a
threshold of $\sim10^6$, the SR concept loses its usefulness, because the atmosphere will
essentially be cleansed of such compounds at the onset of a precipitation event and
subsequent precipitation will simply dilute the concentrations (Lei and Wania, 2004). As
such, measured SRs that are smaller than these very high estimated ones are not too
surprising.
**4.4 Diffusive Air-Water Gas Exchange.** The water-air equilibrium status was evaluated
using fugacity ratios ($f_W/f_A$), whereby $f_W/f_A$ values lower (higher) than 1 indicate a tendency
for net deposition (volatilization). The estimated fugacity ratios for five OPEs (TBP, TCEP,
TCPP, TPhP, and EHDPP) are given in Table S14. This estimation of $f_W/f_A$ incurs substantial
uncertainty because of uncertainty in $K_{AW}$ and the passive sampling rates, and because it
involves combining air and water data obtained during different time periods (Oh et al., 2023;
Zhan et al., 2023). Nevertheless, the $f_W/f_A$ values in BC and QC were so far below unity, that
one can confidently assert that all five OPEs were net deposited from atmosphere to water.
Ma et al. (2021) also reported that almost all five OPEs, except TBP, underwent net gas
phase deposition in the Lower Great Lakes Region.
**4.5 Relative Abundance of OPEs in Different Environmental Media.** The frequent
detection of TCPP, TCEP, TBP, and TPhP in PASs, the gas and particle phase of the AAS, ,
PWSs, and PCPNs in QC and BC allows us to investigate the relative abundance of these
OPEs in different environmental media (Figure 3). Chlorinated compounds (TCPP and
TCEP) were dominant in all environmental media regardless sampling locations, which is
consistent with observations in gaseous and aqueous phases in the Great Lakes region (Ma
et al., 2021). Specifically, TCPP was the most abundant of the four OPEs in all types of
samples, except for the gas phase in Tadoussac and the PWS. By reporting the relative
abundance of the OPEs in PASs separately for industrial, urban, and rural sites, we find a
consistent pattern in both QC and BC, namely that the relative abundance of halogenated
OPEs (TCPP and TCEP) decreased from industrial (65% in QC and 79% in BC) to urban
(59% in QC and 63% in BC) to rural sites (52% in QC and 53% in BC) with a concomitant





increase of two nonhalogenated OPEs (TBP and TPhP). This is consistent with previous
studies (Kurt-Karakus et al., 2018; Zhang et al., 2019), but contrasts with the predominance
of TCPP and TCEP reported for Antarctic air (Wang et al., 2020a). The higher abundance
of TPhP at rural sites would be consistent with a relatively higher long-range transport
potential (LRTP) estimated with the improved OECD Pov and LRTP Screening Tool
(OECD Tool) (Breivik et al., 2022) (Table S15). Even though the observed higher
abundance of TBP in remote areas is inconsistent with its relatively low estimated LRTP,
Sühring et al. (2020) indicated that non-chlorinated OPEs could be subject to LRTP.

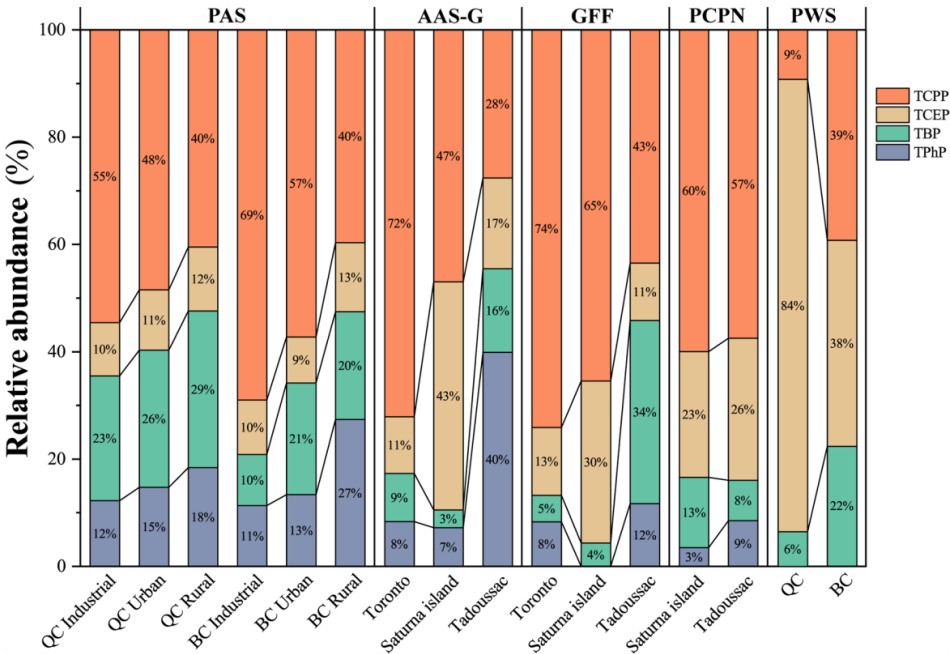


**Figure 3**    The relative abundance of four frequently detected OPEs in passive air samples (PAS),
gas phase active air samples (AAS-G), glass fiber filter samples (GFF), precipitation
samples (PCPN), and passive water samples (PWS) in QC and BC. Due to the large
concentration ranges, geomean was used for calculating the relative abundance for
PASs

The relative abundance of OPEs in the gas and particle phase at the three AAS sites was
similar. Mirroring the observation in the PASs, the relative contribution of chlorinated OPEs
was higher in urban and industrial Toronto than in rural Saturna and remote Tadoussac.
Chen et al. (2019) and Salamova et al. (2013) reported a similar trend for OPEs in dust and
atmospheric particles, respectively. Interestingly, in the atmospheric gas and particle phases,



the higher abundance for TPhP in Tadoussac compared to Toronto suggests that long-range
transport was dominant, despite Toronto being a highly populated city. This observation
aligns with the above findings of a low $\Delta H_{\text{AS-app}}$, the low median concentration level in
Toronto compared to that in Tadoussac (Table S5), as well as the spatial patterns, cluster
analyses, and a weak linear relationship with population previously reported in Li et al.
(submitted). Collectively, these pieces of evidence indicate that TPhP is more closely related
to industrial activities and subject to long-range transport.
The abundance of chlorinated OPEs in PCPN and PWS was much greater than those of the
nonhalogenated OPEs, which is consistent with the dominance of TCPP and TCEP in the
dissolved phase of water sampled from urban and rural watersheds in Toronto (Awonaike et
al., 2021). Based on section 4.4, net diffusion from the atmosphere to water occurred in BC
and QC, therefore, the high abundance of chlorinated OPEs can be explained by their
relatively high $K_{\text{WA}}$ (Tables S4 & S15).
**5. IMPLICATIONS**
Some observations made here are conforming with general expectations regarding the
environmental behaviour of semi-volatile organic chemicals, such as higher gas phase
concentrations and a decrease in the particle bound fraction at higher ambient temperatures.
Also, the measured precipitation scavenging ratios, while high, can be reconciled with
equilibrium partitioning ratios of gaseous OPEs that favour aqueous phases over the gas
phase. Other observations are more puzzling, such as the general lack of a clear relationship
between OPE volatility and the observed gas-particle partitioning behaviour. Furthermore,
the strong temperature dependence of OPE gas phase concentrations that indicates the
importance of temperature-driven local air-surface exchange processes is not entirely
consistent with the low air/water fugacity ratios which suggest that gaseous air-water
exchange is strongly depositional. One possible explanation is that the measured seasonal
concentration variability is less a reflection of temperature driven air-surface exchange and
instead indicates that more OPE enter, or are formed in, the atmosphere in summer. Potential
mechanisms are (i) an increased release of OPEs at higher temperatures from outdoor
materials to which they have been added (Kemmlein et al. 2003), (ii) a faster ventilation of
OPE emitted indoors (Stamp et al., 2022, Han et al., 2024), and (iii) the atmospheric
oxidation of organophosphite precursors (Liu et al., 2023).
Our data also highlight that the understanding of the atmospheric dispersion potential of





OPEs is still incomplete. While a relatively high long range transport potential for aryl-OPE
(TPhP) is consistent with the results from the OECD Tool (Kung et al., 2022; Sühring et al.,
2020), the higher or constant relative abundance for TBP at remote sites does not align with
predictions, which indicate a limited LRTP for TBP. This, too, may be related to the
unpredictable gas-particle partitioning behaviour of the OPEs and the role of gas and multi-
phase transformation processes, e.g. the possibility that TBP originates in part from the
transformation of precursors. More research is needed to better understand the atmospheric
gas-particle partitioning behaviour of the OPEs and to elucidate the role that transformation
reactions may play in this regard.

**Code and data availability**

All data generated for this project are contained in the Supplement.

**Supplement**

The supplement related to this article is available online at: xxxxx.

**Author contributions**

YL, FZ, and JO prepared and extracted the PASs and the Toronto AASs. YL and FZ also
took the Toronto AAS. YDL prepared standards. CS prepared, obtained, and extracted
samples from Saturna Island and Tadoussac as well as the PWSs and analyzed the particle
samples. KL and FAPCG deployed and retrieved PASs and PWSs in British Columbia. ABC,
ZL, HH, FZ, and FW deployed/retrieved PASs and PWSs in Quebec. YL compiled and
interpreted data. YL wrote the manuscript under the guidance of FW with input by the other
co-authors. HH coordinated the project. All authors reviewed the manuscript.

**Competing interests**

The contact author has declared that none of the authors has any competing interests.

**Acknowledgements**

We thank Geri Crooks, Alexandre Costa, Yannick Lapointe, Louis-Georges Esquilat,
Jocelyn Praud, Sandrine Vigneron, François Gagnon, Jonathan Pritchard, Alessia Colussi,
Nicolas Alexandrou, Abigaëlle Dalpé Castilloux, Christian Boutot, Bruno Cayouette, Fella
Moualek, Frédérik Bélanger, Claude Lapierre, Félix Ledoux, Samuel Turgeon, Sarah
Duquette and the CAPMON team for their assistance in deploying samplers and providing
facilities/permissions to the sampling locations.



**Financial support**
This research has been supported by Environment and Climate Change Canada under the
Whale Initiative 1.0 (grants no. GCXE20S008, GCXE20S010, GCXE20S011), and a
Connaught scholarship to Yuening Li.

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
