# Peer review of "Precipitation Scavenging, and Air-Water Equilibrium of"

_EGUsphere, 2024_

## Author Comment (AC1)

**Response to Reviewer 2**

*General comments*

*Review of "Seasonal Air Concentration Variability, Gas/Particle Partitioning, Precipitation Scavenging, and Air-Water Equilibrium of Organophosphate Esters in Southern Canada" by Li et al.*

*Li et al. present extensive long-term measurements of organophosphate esters (OPE) in air, particles, precipitation and bodies of water. Measurements of OPEs across different environments are quite rare and merit publication. The results show OPEs are ubiquitous and show moderate seasonal trends in some cases. In addition, the authors discuss partitioning between these environments and whether they fit their theoretical or estimated partitioning coefficients. The authors did an excellent job discussing the data, citing relevant references and discussing the limitations of these challenging measurements. The article should be accepted after addressing the following small comments*

> We appreciate the positive feedback.

*Specific comments*

*Figures S1 and S2 can be moved to the main text for readability purposes.*

> While we appreciate the suggestion, we still prefer to leave Figures S1 and S2 in the supporting information, considering (i) their large size and (ii) that the data in these two figures are displayed differently in Figure 1.

*Line 201: "We are no comparing..." no should read not.*

> We will correct this typo by replacing "no" with "not".

*Section 3.2: The measured concentrations vary significantly both on a species level as well as a function of location. Could the authors expand upon why some OPEs show higher concentrations in places like Antarctica vs Toronto or similar concentrations in urban and rural environments?*

> OPE concentrations in Antarctica that are higher, or within the same range as, concentrations we measured in Toronto are difficult to explain by long-range atmospheric transport. Local sources could be responsible for the OPE concentrations reported by Wang et al. (2020a), considering that their Antarctic sampling location was close to the Chinese Great Wall station. Sühring et al. (2016) also attributed higher OPE concentrations at Resolute Bay in the Arctic to the influenced of local sources.
>
> Concentration levels of OPEs on atmospheric particles on Saturna Island and in suburban Toronto were similar. We do not have a clear explanation for this observation, although there is the possibility of local sources influencing the deployment location of the active air samplers on Saturna Island. For example, this site was close to a weather station and a ferry terminal.

*Line 277-278: "the dispersion plume of the Montreal waste water treatment plant enters the river at 45 40' N, 73 28' W and stays on the north side of the river" Perhaps this feature can be added to the figure as a marker.*

> While the scale of the map in Figure 1 is too small to indicate this feature graphically, we will move this sentence to the caption of Figure 1.

*Line 368-370: "However, the unexpectedly low fraction observed in the particle phase may suggest that TPhP and EHDPP are emitted at higher temperatures and are not in a state of equilibrium between gas and particle phase" What would prevent these species from achieving equilibrium within the timescales of the measurements?*

> We refer to the study by Zhao et al. (2021a) for a detailed explanation. Zhao et al. (2021a) performed simple model calculations that show that compounds with very low volatility emitted at high temperature do not reach equilibrium because "the time scale for chemical exchange between the gas and particle phases [is] longer than the deposition

removal time scale of the sorbed chemicals". Simply speaking, the particles are not airborne for long enough to reach equilibrium.

Zhao et al. (2021b) had previously observed gas-particle partitioning of OPEs that appears to indicate non-equilibrium. They suggested that this could be a result of the operationally defined gas phase which includes very fine and ultrafine particles that might pass through glass fiber filters. This would cause an overestimation of the fraction of OPEs in the gas phase. We will now add a sentence indicating this as an alternative explanation for the apparent non-equilibrium: "Alternatively, the fraction of TPhP and EHDPP in the gas phase may have been overestimated if very fine and ultrafine particles containing these OPEs passed through the glass fiber filters (Zhao et al., 2021b)."

We note that earlier studies often observed the opposite behaviour, namely much larger particle-bound fractions of relatively volatile OPEs than might be expected.

*Line 461-463: Specific industries that use OPEs could also be more active in the summer months e.g. construction.*

Thanks for pointing this out. We will modify the original sentences by adding the potential influence of seasonally variable industrial activities: "(i) an increased release of OPEs at higher temperatures from outdoor materials to which they have been added (Kemmlein et al. 2003), (ii) a faster ventilation of OPE emitted indoors (Stamp et al., 2022, Han et al., 2024), ), **(iii) more active industrial activities, such as construction, using products containing OPEs in the summer months,** and (iv) the atmospheric oxidation of organophosphite precursors (Liu et al., 2023)."

All references cited in this response can be found in the submitted preprint of the manuscript.

---

## Author Comment (AC2)

**Response to Reviewer 1**

*General comments*

*Organophosphate esters (OPEs) are emerging contaminants that have attracted significant attention due to their negative impact on the environment and human health. While there are numerous reports of the occurrence of OPEs in the atmospheric environment, studies on the gas-particle partitioning and precipitation scavenging of OPEs are rare. In particular, no previous studies have investigated OPEs in different environmental media (atmospheric gas and particle phases, precipitation, and surface water) simultaneously. Based on the comprehensive filed measurements of OPEs in Southern Canada, this study provides new insights into the seasonal variability, gas-particle partitioning behavior, precipitation scavenging, and air-water equilibrium status of OPEs. Such information would be valuable to understand the atmospheric fate of OPEs. Therefore, I recommend publication of this manuscript after minor revisions, as outlined below.*

> We appreciate the reviewer's endorsement of our work.

*Specific comments*

*1. Line 326, unlike TCEPi and TPHPi (which are produced in large quantity), the usage and production of TCPPi have not been reported. As a result, the formation of TCPP from TCPPi seems unlikely.*

> The reviewer is correct that the usage and production of TCPPi have not been reported. We also mentioned this in lines 330 and 331. Nevertheless, the absence of reports does not neccesarily mean that this chemical has not been produced or used. As such production and use at least is possible, we suggested this possibility. More information is needed regarding the possible formation of TCPP from TCPPi.

*2. It is known that OPAs can transform to OPEs through atmospheric reactions. However, it is difficult to evaluate the contribution of OPA transformation chemistry to the measured OPEs in air due to the complex atmospheric processes. The $\Delta H_{AS}$-app analysis in Section 4.1 may provide a potential tool to examine this issue. The authors may want to discuss this point in the manuscript.*

> Thanks for this suggestion. While we agree that the measured $\Delta H_{\text{AS-app}}$ may potentially contain information on the contribution of the transformation of OPAs to the presence of OPEs, it likely would be beset by large uncertainties. The value of $\Delta H_{\text{AS-app}}$ can be influenced by atmospheric advection and by several processes that vary with temperature, such as the OPE source strength to the atmosphere, the exchange between air and surface, and the transformation of OPA to OPEs. If the influence of temperature on OPE source strength is trivial and the value of $\Delta H_{\text{AS-ap}}$ is higher than $\Delta H_{\text{AW}}$ and $\Delta H_{\text{AO}}$, the difference between $\Delta H_{\text{AS-app}}$ and $\Delta H_{\text{AW}}$ or $\Delta H_{\text{AO}}$ may contain information on the contribution of the transformation of OPA. However, this can at most be considered semi-quantitative and will incur high uncertainties. We will add the following sentence to section 4.1:

> "The value of the measured $\Delta H_{\text{AS-app}}$ may potentially contain information on the contribution of the transformation of OPAs to OPEs in the atmosphere, i.e., the extent to which $\Delta H_{\text{AS-app}}$ exceeds $\Delta H_{\text{AW}}$ and $\Delta H_{\text{AO}}$ may indicate the extent of such transformation. However, this would be beset by high uncertainties considering the complex set of factors influencing the $\Delta H_{\text{AS-app}}$"

*3. Line 359, It is surprising that the particle-phase fractions of TCPP and EHDPP in Toronto are 56-68% given their low-volatility nature. How about the measurement results in other urban areas?*

> In our study, Toronto was the only urban area in which we had an active air sampler deployed. The other two active air sampling locations were deployed in remote areas. Wang et al. (2020b) reported particle-bound fractions for TCPP and EHDPP of 40% and 89%, respectively, when conducting active air sampling in an urban area in Dalian, China.

We had referred to the modelling study by Zhao et al. (2021a) to explain why the particle sorption of EHDPP may be lower than expected: "the unexpectedly low fraction observed in the particle phase may suggest that TPhP and EHDPP are emitted at higher temperatures and are not in a state of equilibrium between gas and particle phase (Zhao et al., 2021a)". We now will add another potential explanation: "Alternatively, the fraction of TPhP and EHDPP in the gas phase may have been overestimated if very fine and ultrafine particles containing these OPEs passed through the glass fiber filters (Zhao et al., 2021b)."

*4. Line 370, Please provide some details regarding the impact of particle composition, relative humidity, and degradation on the gas-particle partitioning of OPEs, so that readers can better understand the OPEs' behavior.*

We thanks the reviewer for the suggestion. However, we do not have empirical data on particle composition, relative humidity and possible degradation reactions to aid in the interpretation of our measurement. We had referenced the work by Li et al. (2017b) and Wu et al. (2020) to indicate that other factors may play a role in the gas-particle partitioning of OPEs. We think it would be too speculative to try to explain our observations based on these factors without empirical data. We will rephrase and expand the sentence as follows: " **While it has been suggested that** the composition of the particles (Li et al., 2017b), relative humidity (Li et al., 2017b; Wu et al., 2020), and degradation of OPEs in gas and particle phases may also influence the gas-particle partitioning of OPEs, **we do not have the empirical data to explore the influence of these factors on our measurements**."

*Technical comments*

*Some typos: Line 201, "We are no comparing…"; Line 407, "regardless of".*

We will correct these typos in the manuscript. "no" will be replaced with "not", and "regardless" will be replaced with "regardless of".

All references cited in this response can be found in the submitted preprint of the manuscript.

---

## Author Response (AR2)

1.) On Line 114, you list "Li et al. submitted." What is the status of this article? If it is not published I think you need to remove this and make any important notes from this paper in the SI section. Most journals are not ok these days listing "submitted" for a paper when readers of the paper may not be able to access this data.

> The status of this article is as follows: It has been favorably reviewed after submission to the journal Environmental Toxicology and Chemistry (ET&C) and we have resubmitted a revised version on October 11. The on-line system indicates that the paper is presently "under review", which we assume means that it was send to one or more of the original reviewers to confirm that the revisions are satisfactory. I requested an update from the handling editor last week, but have not received a response. We are reluctant to remove the citations to the submitted papers, because we are confident that the paper will be accepted for publication in ET&C shortly, quite possibly before the ACP manuscript reaches the stage of page proof corrections. We would be agreeable to delay sending the ACP manuscript to the page proof stage until the ET&C submission's acceptance is confirmed.

2.) For the high-volume and low-volume air samplers, I think you need to list the vendor and model of the samplers you are using in the Methods section. This is so future researchers may try to reproduce your measurements.

We have added more detail on the sampling methods, as follows:

> "Twelve 24-hour air samples were collected monthly at a location on Saturna Island, British Columbia (BC) (48.7753N, -123.1283W) between December 2019 and November 2020 and in the vicinity of Tadoussac, Quebec (QC) (48.1415N, -69.6991W) between December 2020 and November 2021 **using high-volume active air samplers (AASs) consisting of a Tisch sampling head (TE-1002-non Teflon with a glass cartridge TE-1009 and a silicone gasket TE-1008-5-Special, Pacwill Environmental, Ontario, Canada) and a high-volume pump (Gast regenerative blower R1102, Cole-Parmer, Illinois, USA)**. Forty-eight consecutive week-long AASs were taken **with the same sampling head assembly and a mid-volume pump (Ametek centrifugal blower DFS 116643-03, RS, Texas, USA)** in the Eastern suburbs of Toronto (43.78371 N, −79.19027 W) (Li et al., 2023a, b, 2024) between June 2020 and May 2021. At all three sites, polyurethane foam (PUF)/XAD/PUF sandwiches and glass-fiber filters (GFFs**, CA28150-214, A/E, 102 mm diameter from VWR**) were used to collect OPEs in the gas and particle phase, respectively. The XAD-2 **was Supelpak™-2 polymeric adsorbent (21130-U, MilliPoreSigma)** and the PUF was a **3-inch TE-1010 (Pacwill Environmental, cut into a 1-inch top PUF and a 2-inch bottom PUF)**. Using **the sampler described by Chan and Perkins (1989)**, precipitation samples (PCPNs) were collected at the AAS sampling locations in BC and QC during the same months as the air samples and the sampling length was ~ 30 days (Oh et al., 2023; Zhan et al., 2023)."

3.) I think it will help some readers of ACP to include a table in the main or SI text that has the chemical structures of the OPEs you targeted in your analyses. For us chemists, I know we can deduce the structures from the chemical names provided, but not all of the readers of this journal are chemists.

> We have added the following figure with chemical structures to the suporting information file: **Figure S1**   Molecular structures of the organophosphate esters (OPEs) targeted in this study. The acronyms are defined in Table S1. Compounds for which labeled internal standards were used are given in bold font.

| Detected OPEs | | Non-detected OPEs | |
| --- | --- | --- | --- |
| Name | Structure | Name | Structure |
| **TEP** |  | TEHP |  |
| **TPrP** |  | ToTP |  |
| **TBP** |  | TmTP |  |
| **TCEP** |  | TpTP |  |
| TCPP |  | T2IPP |  |
| **TDCPP** |  | T35DMPP |  |
| **TPhP** |  | TDBPP |  |

| Injection Standard | |
| --- | --- |
| Triamyl phosphate |  |

**TBEP**

**EHDPP**